# Cross-Cell-Type Prediction of TF-Binding Site by Integrating Convolutional Neural Network and Adversarial Network

**DOI:** 10.3390/ijms20143425

**Published:** 2019-07-12

**Authors:** Gongqiang Lan, Jiyun Zhou, Ruifeng Xu, Qin Lu, Hongpeng Wang

**Affiliations:** 1School of Computer Science and Technology, Harbin Institute of Technology, Shenzhen 518055, China; 2Department of Computing, The Hong Kong Polytechnic University, Hong Kong 810005, China

**Keywords:** TF-binding site, cross-cell-type, deep learning, Convolutional Neural Network, Adversarial Network

## Abstract

Transcription factor binding sites (TFBSs) play an important role in gene expression regulation. Many computational methods for TFBS prediction need sufficient labeled data. However, many transcription factors (TFs) lack labeled data in cell types. We propose a novel method, referred to as DANN_TF, for TFBS prediction. DANN_TF consists of a feature extractor, a label predictor, and a domain classifier. The feature extractor and the domain classifier constitute an Adversarial Network, which ensures that learned features are common features across different cell types. DANN_TF is evaluated on five TFs in five cell types with a total of 25 cell-type TF pairs and compared to a baseline method which does not use Adversarial Network. For both data augmentation and cross-cell-type prediction, DANN_TF performs better than the baseline method on most cell-type TF pairs. DANN_TF is further evaluated by an additional 13 TFs in the five cell types with a total of 65 cell-type TF pairs. Results show that DANN_TF achieves significantly higher AUC than the baseline method on 96.9% pairs of the 65 cell-type TF pairs. This is a strong indication that DANN_TF can indeed learn common features for cross-cell-type TFBS prediction.

## 1. Background

Gene expression contains gene transcription, splicing, translation, and conversion into biologically active protein molecules in vivo. Transcriptional regulation for genes is completed by transcription factors (TFs) binding to specific sites of these genes. These sites are called Transcription Factor Binding Sites (TFBSs). Thus, TFBS prediction is crucial for understanding transcriptional regulatory networks and gene expression regulation [1,2,3].

Both experimental and computational methods were proposed to predict TFBSs. ChIP-chip [4,5] and ChIP-seq are two representative experimental methods. ChIP-chip combines the use of the ChIP technology and the gene chip technology. It first obtains DNA fragments by ChIP experiments and then uses gene chips to obtain DNA fragments that can bind to target TFs. ChIP-seq [5,6,7] was proposed to identify TFBSs at genome scale, which combines the ChIP technology and high-throughput sequencing. Although these experimental methods can identify TFBSs accurately, they are labor-intensive and costly to run. Therefore, it is urgent to propose computational methods for TFBS prediction.

Many computational methods were proposed to predict TFBSs. As TFBSs are degenerate sequence motifs [8], position weight matrices (PWMs) [9] are used to represent TFBSs by many computational methods. PWM is derived from a set of aligned sequences which have related biology functions. PWMs of TFs can be represented by matrices, whose elements are the binding preferences of TFs at each position. As PWM can show base variations of each position of TFBSs, it is a popular method for DNA motif representations [9]. As PWMs of many TFs can be retrieved from two transcription factor databases: JASPAR [10] and TRANSFAC [11], recent works attempted to predict TFBSs for those TFs by simply scanning the genome using their PWMs. For example, Comet [12] and ModuleMiner [13] used PWMs documented in these two databases to identify TFBSs. However, these prediction methods always generate too many false positives due to the factor that PWM cannot extract position dependencies between positions.

In addition to the use of PWMs, many methods also used machine learning to predict TFBSs. Talebzadeh and Zare-Mirakabad [14] developed a method by the combined use of two types of features: the closest distance to the nearest histone and the total number of histone modifications. Won et al. [15] developed an integrated method (called Chromia) using PWM and histone modification features. Results showed that Chromia significantly outperforms many available methods on 13 TFs in the mouse embryonic stem (mES) cell. As Support Vector Machine (SVM) has been successfully applied to address many prediction problems in Bioinformtics, some works also used SVM to predict TFBSs of TFs. For example, Kumar et al. [16] used SVM to predict site occupancy and binding strength. In addition to SVM, random forest [17] was also used to predict TFBSs. For example, Tsai et al. [18] used random forest to examine the relative importance of sequence features, histone modification features, as well as DNA structure features in prediction and indicated that all the three feature types are significant in TFBS prediction.

Recent works suggested that deep learning methods are useful for TFBS prediction. DeepBind [19], DeepSea [20] and DanQ [21] are three representative deep learning methods. DeepBind [19], proposed by Alipanahi et al., used Convolutional Neural Network (CNN) to learn representations for TFBSs from their DNA sequences. DeepSea [20], proposed by Zhou and Troyanskaya, combined the use of CNN and multi-task learning to learn representations for DNA sequences. DanQ [21], an improved model of DeepSEA proposed by Quang and Xie, combined the use of CNN and Recurrent neural network (RNN). Both DeepSEA and DanQ used multi-task learning to learn representations for TFBSs. They include 690 TFBS prediction tasks, 104 histone modification prediction tasks, 125 prediction tasks for DNase I-hypersensitive sites (DHSs). These three deep learning methods achieve very good performance and are considered the state-of-the-art works.

When there exists sufficient labeled data, most existing methods can achieve very good performance. However, labeled data for most TFs in a lot of cell types can only be obtained by ChIP-chip or ChIP-seq, which are labor-intensive and costly to run. Thus, TFs do not have sufficient labeled data in many cell types, and some do not have any labeled data. It is quite challenging to predict TFBSs for TFs in cell types which lack labeled data. Nevertheless, several studies have shown that TFBSs of a TF in different cell types have common histone modifications and have common binding motifs. Therefore, computational methods can leverage on labeled data of target TFs available in cross-cell-type to predict TFBSs in cell types that lack labeled data. In this paper, we propose a cross-cell-type method, referred to as DANN_TF (see Materials and Methods), by combining the use of CNN and Adversarial Network [22] to learn common features among multiple cell types with available labeled data. The learned common features are aimed to predict TFBSs for cell types that do not have sufficient labeled data or do not have any labeled data. The resource and executable code are freely available at http://hlt.hitsz.edu.cn/DANNTF/ and http://www.hitsz-hlt.com:8080/DANNTF/.

## 2. Experiments and Results

### 2.1. Experimental Settings

To evaluate DANN_TF at genomic scale, we predict TFBSs for the five TFs in the five cell types (see Materials and Methods, Table 5) with a total of 25 prediction tasks. When the target cell type has labeled data, DANN_TF amounts to a data augmentation method due to the fact that labeled data in the target cell type is augmented by labeled data in source cell types. To evaluate the influence of the size of labeled data in the target cell type on DANN_TF, we evaluate the performance of semi-supervised prediction by DANN_TF. In semi-supervised prediction, we conduct three experiments, in which 50%, 20% and 10% trained data in the target cell type are labeled, respectively. When the target cell type has no labeled data, DANN_TF amounts to a cross-cell-type method due to the fact that labeled data in source cell types is used to predict TFBSs for the target cell type. In each evaluation, our proposed DANN_TF is compared with a baseline method. This baseline method is similar to DANN_TF except that it does not use Adversarial Network in prediction. To further evaluate the validity of labeled data available in source cell types for prediction in the target cell type, we compare our DANN_TF with supervised prediction by the baseline method. Finally, we compare our proposed DANN_TF with existing methods with state-of-the-art performance. In these evaluations, labeled data of each TF in each cell type are divided into 10 separate folds of equal size: 8 folds for training, 1 fold for validation, and 1 fold for test. The above process is repeated for ten times until each fold is tested once. Finally, the performance on the 10 folds are averaged.

The inputs of the feature extractor (see Materials and Methods, Figure 6) are feature matrices with dimension of N×L=4×101. The feature extractor in DANN_TF uses two convolution layers with $N_{1}=32$ and $N_{2}=48$ kernels, respectively, and each is followed by a max pooling layer. The kernel size of the two convolution layers and the pooling size of the two pooling layers are M=5. The stride of the two convolution layers and the two pooling layers are 1 and 2, respectively. A dropout regularization layer with dropout probability of 0.7 is used to avoid overfitting. The label predictor has two fully connected layers of 100 neurons followed by a softmax classifier. The domain classifier has a fully connected layer with 100 neurons followed by a softmax classifier. The domain adaptation parameter α is set to 1. Our DANN_TF model was trained using the Momentum algorithm [23] with batch size of 128 instances. The momentum and the learning rate are set as 0.9 and 0.001, respectively. DANN_TF was implemented using the tensorflow-gpu 1.0 library.

### 2.2. Evaluation Metrics

We evaluate our proposed DANN_TF and compare with the baseline method using AUC. AUC [24] is defined as the area under the Receiver Operating Characteristic (ROC) curve. ROC plots the true positive rate (sensitivity) versus the false negative rate (1-specificity) of different thresholds on the importance score. AUC measures the similarity of predictions to the known gold standard and the value is equivalent to the probability that a randomly chosen positive instance is ranked higher than a randomly chosen negative instance. An AUC of 1.0 and 0.5 indicate the best performance and the random performance, respectively.

A recent work [25] suggested that F1 score is another useful evaluation metric for TFBS prediction problem.
(1)$$\begin{array}{c} F1=2\times P\times R\;/\;(P+R))\;,\\ P=TP\;/\;(TP+FP)\;,\\ R=TP\;/\;(TP+FN)\;, \end{array}$$
where TP denotes the number of true positives, FP denotes the number of false positives and FN denotes the number of false negatives. Moreover, we used *p* value, calculated by the Wilcoxon signed-ranks test [26], to evaluate whether performance differences in this paper are significant.

### 2.3. The Impact of Negative Samples on Experiment

In general, there are two negative sequence generation methods: random method and shuffle method. In the shuffle method, a non-TFBS was constructed for every TFBS by shuffling dinucleotides in this TFBS to keep the dinucleotide distribution unchanged. In the random method, non-TFBSs are random genomic loci that are not annotated as TFBS. To evaluate the impacts of the two methods on the performance of our proposed DANN_TF, we evaluate the performance of DANN_TF on JunD in the five cell types for cross-cell-type prediction by using negative sequences generated by the two negative sequence generation methods. For JunD in each cell type, we first used the shuffle method and the random method to generate 15 sets of non-TFBSs randomly, respectively. All these 30 sets contain equal number of non-TFBSs. Next, we obtain 30 data sets by combining the 30 sets of non-TFBSs with all the TFBSs. Finally, we trained our proposed DANN_TF by combining the unlabeled training data of the target cell type and labeled data of source cell types and trained the baseline method by labeled training data of source cell types. The performance of DANN_TF trained by different sets of non-TFBSs are listed in Appendix A. Results show that the performance of the shuffle method and the random method are almost the same. Therefore, it is a strong indication that the two methods have little effect on the performance of DANN_TF.

### 2.4. Results of Data Augmentation by DANN_TF

To evaluate the performance of data augmentation by DANN_TF, our proposed DANN_TF and the baseline method are trained by the training data of the target cell type and all labeled data of source cell types. They are validated and tested by the validation data and the test data of the target cell type, respectively.

Results of DANN_TF in Figure 1a,b show that DANN_TF achieves higher AUC and F1 score than the baseline method for most cell-type TF pairs. A cell-type TF pair denotes a prediction task.

Results in Figure 1c,d show that the first quartile, the median and the third quartile of AUC and F1 score of DANN_TF are higher than that of the baseline method. Details of AUC and F1 score of DANN_TF and the baseline method on the 25 cell-type TF pairs is shown in Appendix A, where the best performers are marked by bold. Results show that DANN_TF achieves higher AUC than the baseline method significantly for 21 pairs out of the 25 cell-type TF pairs. The maximum improvement and the average improvement are 9.19% and 0.95%, respectively. DANN_TF also achieves higher F1 score than the baseline method significantly for 23 pairs out of the 25 cell-type TF pairs. The maximum improvement and the average improvement are 16.29% and 1.47%, respectively. DANN_TF achieves the maximum improvement when predicting TFBSs for JunD in GM12878. One possible reason is that labeled training data of JunD in GM12878 is less than other cell-type TF pairs and DANN_TF can leverage labeled data available in the source cell types to greatly improve the performance for GM12878.

Most noticeably, our proposed DANN_TF performs better than the baseline method with a larger margin for JunD in three cell types. The improvements in GM12878, H1-hESC and K562 are 9% in AUC, 4% in AUC, and 2.5% in F1 score, respectively. For REST, the improvement is more than 1.5% in AUC for GM12878 and more than 1% in F1 score for three cell types. For GABPA and USF2, although their improvements are smaller than that of the other three TFs, the improvement in F1 score for some cell types are also more than 1%. This is a strong indication that Adversarial Network indeed play an important role in learning common features among the target cell type and source cell types. Exceptionally, DANN_TF achieves lower AUC than the baseline method for CTCF in four cell types. One possible reason is that the labeled training data of CTCF in these four cell types are sufficient and augmenting their training data by labeled data available in other cell types may bring a lot of noise.

### 2.5. Results of Semi-Supervised Prediction by DANN_TF

To evaluate the performance of semi-supervised prediction by DANN_TF, we suppose that only a portion of the training data of the target cell type is labeled and the remaining training data is unlabeled. To evaluate the influence of the number of the labeled training data in the target cell type on DANN_TF, three experiments are conducted: (1) 50% training data is labeled, (2) 20% training data is labeled and (3) 10% training data is labeled.

Results of the three experiments in Figure 2a–f show that DANN_TF performs better than the baseline method in both AUC and F1 score for all the 25 cell-type TF pairs in all the three experiments. Details of AUC and F1 score of DANN_TF and the baseline method for each cell-type TF pair in the three experiments is listed in Appendix A, respectively. For the experiment with 50% labeled training data, the maximum improvement and the average improvement in AUC are 8.69% and 1.14%, respectively. The maximum improvement and the average improvement in F1 score are 12.32% and 1.35%, respectively. The improvements in AUC on JunD in GM12878 and H1-hESC as well as USF2 in H1-hESC are 8.69%, 5.1% and 2.41%, respectively. For the experiment with 20% labeled training data, the maximum improvement and the average improvement in AUC are 8.71% and 1.5%, respectively. The maximum improvement and the average improvement in F1 score are 10.46% and 1.75%, respectively. The improvements in AUC on JunD in GM12878 and H1-hESC is 8.71% and 4.19%, respectively. The improvements in AUC on REST in GM12878 and USF2 in H1-hESC are 2.56% and 2.76%, respectively. For the experiment with 10% labeled training data, the maximum improvement and the average improvement in AUC are 8.66% and 1.06%, respectively. The maximum improvement and the average improvement in F1 score are 10.64% and 1.77%, respectively. The improvements in AUC on JunD in GM12878 and H1-hESC is 9.06% and 3.59%, respectively. The improvements in AUC on REST in GM12878 and USF2 in H1-hESC are 2.22% and 2.61%, respectively.

The analysis of the three experiments shows that DANN_TF achieves high performance when labeled training data is reduced. On the contrary, the performance of the baseline method is reduced obviously when labeled training data is reduced. This is a strong indication that Adversarial Network indeed can help to learn common features among the target cell type and source cell types when the number of labeled training data in the target cell type is limited.

### 2.6. Results of Cross-Cell-Type Prediction by DANN_TF

To evaluate the performance of cross-cell-type prediction by our proposed DANN_TF, we suppose that all the training data of the target cell type is unlabeled while the validation data as well as the test data are labeled. Thus, our proposed DANN_TF is trained by combining unlabeled training data of the target cell type and labeled data of source cell types. As the baseline method cannot use unlabeled data, the baseline method is trained by labeled data of source cell types. They are validated and tested by the validation data and the test data of the target cell type, respectively. As our goal is to predict TFBSs of TFs in the target cell type, and DANN_TF is trained by unlabeled training data of the target cell type and labeled data of source cell types, thus DANN_TF is an unsupervised method. For example, when we use DANN_TF to predict TFBSs for CTCF in GM1278, the training data of DANN_TF are the combination of unlabeled training data of CTCF in GM12878 and labeled data of CTCT in other four cell types. As DANN_TF do not use any labeled training data of CTCF in GM12878, predictions of DANN_TF for CTCF in GM12878 are unsupervised predictions.

The comparison listed in Figure 3a,b shows that DANN_TF performs better than the baseline method for most cell-type TF pairs. Figure 3c,d show that the first quartile, the median and the third quartile of the performance of DANN_TF are higher than that of the baseline method. Details of AUC and F1 score of DANN_TF and the baseline method for each cell-type TF pair is listed in Appendix A, where the best performers are marked by bold. The table shows that DANN_TF performs better than the baseline method significantly on all the 25 cell-type TF pairs in both AUC and F1 score. More specifically, for JunD in GM12878 and H1-hESC, REST in GM12878 as well as USF2 in H1-hESC, the improvements are more than 1% in AUC significantly. In terms of F1 score, DANN_TF performs better than the baseline method significantly for most cell-type TF pairs. These results indicate that our proposed DANN_TF can achieve better performance than the baseline method for cross-cell-type prediction.

### 2.7. Comparison between Cross-Cell-Type Prediction by DANN_TF and Supervised Prediction by the Baseline Method

To further evaluate the performance of cross-cell-type prediction by our proposed DANN_TF, we compare the performance of cross-cell-type prediction by DANN_TF to that of supervised prediction by the baseline method. Our proposed DANN_TF is trained by the combined use of unlabeled training data of the target cell type and labeled data of source cell types. The baseline method is trained by the training data of the target cell type, which are labeled.

The performance of cross-cell-type prediction and supervised prediction are listed in Table 1. The table shows that there are 24 pairs out of the 25 cell-type TF pairs on which DANN_TF performs better than the baseline method significantly in AUC. For F1 score, there are 23 pairs on which DANN_TF performs better than the baseline method significantly. The average improvement is more than 5% in AUC and more than 8% in F1 score, which is a very prominent improvement. More specifically, for JunD, REST, and GABPA, our proposed DANN_TF performs better than the baseline method with a larger margin for almost all the five cell types. DANN_TF performs better than the baseline method by more than 17% AUC for REST in GM12878, more than 9% AUC for REST in H1-hESC. Meanwhile, for CTCF and USF2, F1 score and AUC in some cell types are also improved by more than 1%. These comparisons show that DANN_TF achieves better performance than the baseline method.

### 2.8. Results for Cross-Cell-Type Prediction by DANN_TF on 13 TFs of the Five Cell Types

We further evaluate DANN_TF for cross-cell-type prediction by additional 13 TFs in the five cell types (see Materials and Methods, Table 6). We suppose that all the training data of the target cell type is unlabeled while the validation data as well as the test data are labeled. Thus, DANN_TF is trained by combining unlabeled training data of the target cell type and labeled data of all the source cell types. As the baseline method cannot be trained by unlabeled data, it is trained by only labeled data of the source cell types. We also compare cross-cell-type prediction by DANN_TF to supervised prediction by the baseline method, where the baseline method is trained by labeled data of the target cell type.

The comparison between cross-cell-type prediction by the baseline method and our proposed DANN_TF and supervised prediction by the baseline method is shown in Figure 4. Figure 4a–d show that DANN_TF achieves higher AUC and F1 score than the baseline method for cross-cell-type prediction, and performs better than supervised prediction for most cell-type TF pairs. Figure 4e,f show that the first quartile, the median and the third quartile of AUC and F1 score for DANN_TF are highest, respectively. Details of AUC and F1 score for these three models on the additional 13 TFs are listed in Appendix A. AUCs and F1 scores of cross-cell-type prediction by DANN_TF and the baseline method and supervised prediction by the baseline method for the 13 TFs in the five cell types are shown in Appendix A. Table 2 and Table 3 summarize AUC gains of DANN_TF compared to the baseline method and supervised prediction, respectively. For the five cell types, DANN_TF outperforms the baseline method on at least 86.4% TFs of all the cell types. The maximum improvement and the average improvement are at least 2.6% and at least 1.5%, re1spectively. The micro average of the maximum improvements and the average improvements are 4.4% and 1.8%, respectively. Moreover, DANN_TF outperforms supervised prediction on at least 86.4% TFs of all the cell types. The maximum improvement and the average improvement are at least 15.0% and at least 6.6%, respectively. The micro average of the maximum improvements and the average improvements are 20.6% and 7.7%, respectively. The F1 score gain of DANN_TF compared to the baseline method and supervised prediction are listed in Appendix A, respectively.

In the 13 TFs, five TFs do not have specific binding motifs in any of the three common databases (JASPAR [10], TRANSFAC [11] and Uniprobe [27]). As these TFs do not have specific binding motifs which can be learned by CNN from labeled data in cross-cell-type, they may achieve low improvements than the sequence-specific TFs. Results listed in Appendix A show that DANN_TF achieves higher AUC than the baseline method and supervised prediction by 1.6% and 7.2% on average, respectively, for the TFs without specific binding motifs. For the sequence-specific TFs, DANN_TF achieves higher AUC than the baseline method and supervised prediction by 2.0% and 8.0% on average, respectively. Although DANN_TF achieves lower improvements for the TFs without specific binding motifs than the sequence-specific TFs, DANN_TF also achieves prominent improvements for them. It indicates that DANN_TF can learn binding features from labeled data in cross-cell-type for the TFs which do not have specific binding motifs.

### 2.9. Comparison between Data Augmentation, Semi-Supervised, and Cross-Cell-Type Prediction

We further compare the performance of cross-cell-type prediction by our proposed DANN_TF with that of data augmentation by the baseline method. Results show that for CTCF in all the five cell types, GABPA in GM12878 and K562, JunD in GM12878, and HeLa-S3, REST in H1-hESC and HepG2 as well as USF2 in H1-hESC, the performance of cross-cell-type prediction by DANN_TF is better than that of data augmentation by the baseline method. Most noticeably, the performance of cross-cell-type prediction by DANN_TF for JunD in GM12878 is better than that of the baseline method for data augmentation by more than 12%. It indicates that DANN_TF trained by labeled data from source cell types can achieve better performance than the baseline method trained by labeled data from both the target cell type and source cell types.

We also compare the performance of data augmentation, semi-supervised, and cross-cell-type prediction by DANN_TF, which is show in Appendix A, and Figure 5. In Figure 5, the horizontal ordinate denotes the 25 cell-type TF pairs. Figure 5a,b show the AUC comparison and the F1 score comparison, respectively.

Results show that the discrepancies of both AUC and F1 score among data augmentation, semi-supervised prediction, and cross-cell-type prediction are very small for most cell-type TF pairs. It indicates that DANN_TF is applicable to TFs in cell types which have insufficient labeled data or do not have any labeled data.

### 2.10. Comparison with the State-of-the-Art Methods

Several deep learning methods including DeepSEA [20] and DanQ [21] achieved state-of-the-art performance. Quang and Xie (2016) also proposed an alternative model of DanQ, called DanQ-JASPAR (DanQ-J), by initializing half of CNN kernels with motifs from JASPAR [28]. We also consider an alternative model of DeepSEA, referred to as DeepSEA-JASPAR (DeepSEA-J), by using the same kernel initializing method as DanQ-JASPAR. We downloaded the torch implementation of DeepSEA [20] from the software’s webpage (http://DeepSEA.princeton.edu/) and the Keras implementation of DanQ and DanQ-JASPAR [21] from the software’s webpage (http://github.com/uci-cbcl/DanQ). In this section, we compare our proposed DANN_TF with DeepSEA, DeepSEA-JASPAR, DanQ, and DanQ-JASPAR on the 25 cell-type TF pairs by ten-fold cross-validation. All these four comparing methods are multi-task learning methods. As the comparison between DANN_TF and them is conducted by 25 cell-type TF pairs, they contain 25 TFBS prediction tasks.

The performance of the four state-of-the-art methods and our proposed DANN_TF is listed in Table 4, where the best performers and the second-best performers are marked by bold and underline, respectively. Please note that DANN_TF is trained by unlabeled training data of the target cell type and labeled data of source cell types while the four state-of-the-art methods are trained by labeled training data of the target cell type and labeled data of source cell types by using their default setup and parameters. The table shows that our proposed DANN_TF performs better than the four state-of-the-art methods on 24 pairs out of the 25 cell-type TF pairs. The maximum and the minimum improvement in AUC are 29.7% and 1.7%, respectively. The average improvement in AUC is 14.8%, which is a very prominent improvement. It manifests that our proposed DANN_TF performs better than the four state-of-the-art methods.

To evaluate the efficiency of our proposed DANN_TF, we compare the training time of DANN_TF for each epoch to that of the baseline method and the four state-of-the-art methods. JunD in the five cell types are taken as examples to evaluate their efficiency. All the methods are trained on NVIDIA GeForce RTX 2080Ti. Results show that DANN_TF takes 45s, 42s, 44s, 42s, and 45s for GM12878, H1-hESC, HeLa-S3, HepG2, and K562, respectively, and the training time of the baseline method for these five cell types are 33s, 34s, 32s, 28s, and 29s, respectively. The training time of DanQ, DanQ-JASPAR, DeepSEA, and DeepSEA-JASPAR are 401s, 355s, 96s, and 85s, respectively. As their predictions for the five cell types are completed by a single model, their average training time for each cell type are 80s, 71s, 19s, and 17s, respectively. In summary, DANN_TF takes less time than DanQ and DanQ-JASPAR and spent a little more time than the baseline method, DeepSEA, and DeepSEA-JASPAR.

## 3. Discussion

TFBS prediction is pivotal for understanding gene expression regulation. Although many computational methods were proposed for TFBS prediction, most existing methods are inapplicable to TFs in cell types which have insufficient labeled data or do not have any labeled data. In this study, we proposed a novel prediction method, referred to as DANN_TF, by using Adversarial Network. DANN_TF aims to make use of common features across different cell types to predict TFBSs for TFs in cell types which lack labeled data. The performance gain compared to the baseline method on five TFs in five cell types shows that Adversarial Network indeed can help learning common features across different cell types. The performance of cross-cell-type prediction by DANN_TF is even better than that of supervised prediction by the baseline method. This is a clear indication that common features learned by DANN_TF from labeled data available in source cell types are indeed useful for cross-cell-type prediction. The performance gain of DANN_TF for cross-cell-type prediction compared to the baseline method and supervised prediction on additional 13 TFs shows that DANN_TF has a good generalization ability. The comparison of DANN_TF to four state-of-the-art methods shows that DANN_TF performs better than them for 24 pairs of the 25 cell-type TF pairs. As TFBSs can promote the research on gene expression regulation, accurate cross-cell-type TFBS prediction can assist people to understand biology functions of rarely studied cell types. Although DANN_TF can leverage labeled data available in source cell types to predict TFBSs for the target cell type which lacks labeled data, it treats labeled data of different cell types equally. Actually, different source cell types have different functional relativity with the target cell type. Therefore, our future works will explore the relative contributions of different source cell types to prediction in the target cell type.

## 4. Materials and Methods

As shown by many recently published works [29,30,31,32], a complete prediction model in bioinformatics should contain the following five components: validation benchmark dataset(s), an effective feature extraction procedure, an efficient predicting algorithm, a set of fair evaluation criteria and comparisons with state-of-the-art methods. The latter two components have been introduced in Experiments and Results. In this section, the definition of TFBS for our prediction task will be introduced first. Then, details of the first three components of our proposed DANN_TF will be described in sequence.

### 4.1. TF-Binding Site (TFBS)

Recently, most studies used ChIP-seq to obtain TFBSs and non-TFBSs. ChIP-seq first provides a signal for each fragment in a genome. Then a peak calling method is applied to identify peaks from the genome according to the given signals. Based on the works by Alipanahi et al. [19] and Zeng et al. [33], the TFBS at each peak is defined as a 101 bp sequence by taking the midpoint of the peak as the center. For example, given a genome *G* with length of *L* (L>>101)
(2)$$G=N_{1}N_{2}N_{3}N_{4}N_{5}N_{6}\cdots N_{i-1}N_{i}N_{i+1}\cdots N_{L}\;,$$
where $N_{1}$ represents the first nucleotide, $N_{2}$ represents the second nucleotide and so forth. For a peak with the midpoint at the position *i* in *G*, its TFBS can be represented as

(3)$$T_{i}=N_{i-50}N_{i-49}{\cdots N}_{i-1}N_{i}N_{i+1}{\cdots N}_{i+49}N_{i+50}\;,$$

Apart from TFBSs, all other nonoverlapping 101-bp DNA fragments in the genome are defined as non-TFBSs.

### 4.2. Datasets

Two data sets are used to evaluate the performance of our proposed DANN_TF. The first one consists of five TFs and the other one contains 13 TFs.

In the first data set, five different cell types are selected: (1) GM12878, a kind of Lymphocyte in humans, (2) H1-hESC, human embryonic stem cells, (3) HeLa-S3, human cervical cancer cells, (4) HepG2, which is derived from a 15-year-old Caucasian liver tissue, and (5) K562, which is the first artificially cultured cells of human myeloid leukemia. The five TFs in this data set contain: (1) CTCF, (2) JunD, (3) GABPA, (4) REST, and (5) USF2. Peaks of these five TFs in the five cell types can be downloaded from ENCODE [34] freely. Peaks can be provided in one of two formats: narrow peak and broad peak. Some TFs have well defined binding sites and can be modeled by narrow peaks while binding sites of other TFs are less well localized and would better be modeled by broader peaks. Thus, the narrow peak format is used if available. Otherwise, the broad peak format is used. The number of TFBSs for the five TFs in the five cell types is listed in Table 5. In the second data set, a total of 13 TFs are used to evaluate the generalization ability of our proposed DANN_TF. These 13 TFs have available TFBSs in all the five cell types. The TFBSs of these TFs in the five cell types were identified by TF ChIP-seq experiments and their peaks can be downloaded from ENCODE. Among the 13 TFs, eight TFs are sequence-specific and have specific binding motifs [10,11,27]. The remaining five TFs (CHD2, EZH2, NRSF, RAD21 and TAF1) do not have specific binding motifs in any of the three common databases (JASPAR [10], TRANSFAC [11] and Uniprobe [27]). The number of TFBSs for the 13 TFs in the five cell types is listed in Table 6.

In contrast to TFBSs, non-TFBSs of a TF are defined as DNA regions which cannot be bound by the TF. Much literature [16] has used a shuffle method to construct non-TFBSs. In the shuffle method, a non-TFBS was constructed for every TFBS by shuffling dinucleotides in this TFBS to keep the dinucleotide distribution unchanged. This way, we can construct the same number of non-TFBSs as TFBSs for each TF. The TFBSs and non-TFBSs for all these cell-type TF pairs are freely available at our webserver.

### 4.3. Feature Representation

The one-hot encoding is commonly used embedding method [35,36]. This method represents each word as a feature vector with the dimension as the vocabulary size. In each vector, only one element is 1, which indicates the current word, and all the remaining elements are 0. DNA is composed by four nucleotide types (A, G, C, T), so the dimension of the one-hot vectors of the four nucleotide types is four. They can be encoded as follows

(4)$$\begin{array}{c} A={\left(0,0,0,1\right)}^{T}\;,\\ C={\left(0,0,1,0\right)}^{T}\;,\\ G={\left(0,1,0,0\right)}^{T}\;,\\ T={\left(1,0,0,0\right)}^{T}\;. \end{array}$$

As both TFBSs and non-TFBSs are composed by 101 nucleotides, they can be encoded by matrices of dimension 4×101. Thus, for a TFBS $T_{i}$, denoted by $T_{i}=AGCTG\cdots CTCA$, its encoded vector can be denoted as

(5)$${\left[\begin{array}{cccccccccc} 0 & 0 & 0 & 1 & 0 & \cdots & 0 & 1 & 0 & 0\\ 0 & 1 & 0 & 0 & 1 & \cdots & 0 & 0 & 0 & 0\\ 0 & 0 & 1 & 0 & 0 & \cdots & 1 & 0 & 1 & 0\\ 1 & 0 & 0 & 0 & 0 & \cdots & 0 & 0 & 0 & 1 \end{array}\right]}_{4\times 101}$$

### 4.4. Transfer Learning

Before introducing our proposed method, we first introduce several related concepts of transfer learning.

Transfer learning [37] is a common problem in machine learning. Transfer learning aims to address one problem by using knowledge learned from a different but related problem. The mathematical definition of transfer learning is formulated as follows: given a source domain $D_{S}=\left\{X_{S},f_{S}\left(X\right)\right\}$ and its learning task $T_{S}$ as well as a target domain $D_{T}=\left\{X_{T},f_{T}\left(X\right)\right\}$ and its learning task $T_{T}$, transfer learning aims to learn the task $T_{T}$ of $D_{T}$ by using the knowledge in $D_{S}$ and $T_{S}$, where $D_{S}\neq D_{T}$, or $T_{S}\neq T_{T}$. In TFBS prediction for a TF in a target cell type, the target cell type is the target domain while other cell types are the source domain, called source cell types. Thus, transfer learning aims to address the lack of labeled data in the target cell type by using labeled data available in source cell types. In our study, we attempt to use Adversarial Network to address transfer learning problem. Adversarial Network was proposed by Goodfellow [38] and achieved great success for deep learning methods. This model contains two components: a discriminator and a generator. The discriminator is used to distinguish the target cell type from source cell types while the generator aims to extract common features among the target cell type and source cell types. As Adversarial Network can help to learn common features for multiple cell types, it is suitable to address transfer learning problems in TFBS prediction.

### 4.5. DANN_TF

In this work, we propose a novel prediction method, referred to as Domain-Adversarial Neural Networks for TF-binding site prediction (DANN_TF), for TFs in cell types which lack labeled data. DANN_TF combines the use of CNN and Adversarial Network to learn common features among the target cell type and source cell types. Then learned common features are aimed to predict TFBSs for TFs in the target cell type. The framework in Figure 6 shows that our proposed DANN_TF contains three components: a feature extractor, a label predictor, and a domain classifier. The green circles denote a input sequence, the blue circles denote learned features by the feature extractor, the dark yellow circle and the pale yellow circle denote TFBS and non-TFBS, respectively, and the dark black circle and the pale black circle denote the target cell type and source cell types, respectively. The input to DANN_TF is a sample of the target cell-type or source cell types. For example, when we use DANN_TF to predict TFBSs of CTCF in GM1278, then GM12878 is the target cell type while the other four cell types are the source cell types. The input to DANN_TF is a TFBS of CTCF in GM12878 or in the other four cell types. The output contains two components: the Label Predictor predicts the input sequence into TFBS or non-TFBS while the Domain Classifier predicts whether the input sequence belongs to the target cell type or the source cell types.

The feature extractor is used to learn common features for the target cell type and source cell types. This feature extractor consists of two convolution layers and each is followed by a max pooling layer. The label predictor outputs prediction results based on features learned by the feature extractor. This label predictor consists of two fully connected layers followed by a softmax classifier. The domain classifier distinguishes whether an input sequence belongs to the target cell type or source cell types based on features learned by the feature extractor. This domain classifier consists of a Gradient Reversal layer (GRL) followed by a fully connected layer and a softmax classifier. The feature extractor acts as a generator while the domain classifier acts as a discriminator, so they constitute an Adversarial Network. DANN_TF provides a real-valued score for a given DNA sequence $T_{i}$ as follows
(6)$$f\left(T_{i}\right)=G_{l}\left(G_{f}\left(T_{i}\right)\right)$$
where $G_{l}\left(\cdot \right)$ and $G_{f}\left(\cdot \right)$ denote the feature extractor and the label predictor in DANN_TF, respectively. The domain classifier $G_{d}\left(\cdot \right)$ plays a role in the real-valued score by constituting an Adversarial Network with the feature extractor $G_{f}\left(\cdot \right)$. Details of these three modules are given in the following sections.

#### 4.5.1. Feature Extractor

The feature extractor $G_{f}\left(\cdot \right)$ plays a role as a generator in DANN_TF. It contains the sequential alternation between two convolution and two pooling layers to learn sequence features at different spatial scales.

For the convolution layers, the input is a matrix *X* of dimension N×L, where *N* is the dimension of each element and *L* is the length of input sequences. The output of the convolution layers is
(7)$$Y_{i,k}=conv_{i,k}\left(X\right)=ReLU\left(\sum_{m=0}^{M-1}\sum_{n=0}^{N-1}W_{mn}^{k}X_{i+m,n}\right)$$
where *i* is the index of an output position and *k* is the index of a kernel. Each convolution kernel $W_{k}$ is an M×N weight matrix with *M* being the window size and *N* being the number of input channels (for the first convolution layer *N* equals 4, for higher-level convolution layers *N* equals the number of kernels in the previous convolution layer). ReLU represents the rectified linear function. The pooling layers are fed with matrices *Y* outputted by the convolution layers and output vectors Z=pooling(Y) with dimension of *d*. An element $Z_{i,k}\left(1\;\leq k\;\leq \;d\right)$ of vector *Z* is computed as
(8)$$pooling_{i,k}\left(Y\right)=max\left(Y_{i\times M,k},Y_{(i\times M+1),k},\ldots ,Y_{(i\times M+M-1),k}\right)$$
where *Y* is the input, *i* is the index for an output position, *k* is the index of a kernel and *M* is the pooling window size.

In summary, the output of the feature extractor is
(9)Z=pooling(conv(pooling(conv(X))))
where conv(·) and pooling(·) denote a convolution layer and a pooling layer, respectively.

#### 4.5.2. Label Predictor

The label predictor is aimed to predict whether input DNA sequences are TFBSs or not based on features learned by the feature extractor. This predictor consists of two fully connected layers followed by a softmax classifier. The label predictor is fed by feature vectors Z, and computes
(10)$$G_{l}\left(Z\right)=softmax(FC\left(FC\left(Z\right)\right)$$
where FC(·) denotes a fully connected layer and softmax(·) denotes the softmax classifier. The softmax classifier contains two output units denoting TFBS and non-TFBS, respectively. To avoid overfitting, a dropout layer is added before the first fully connected layer. With the dropout technique, any entry is set to 0 randomly with a dropout rate.

#### 4.5.3. Domain Classifier

The domain classifier aims to discriminate DNA sequences into the target cell type or source cell types. This classifier consists of a GRL followed by a fully connected layer and a softmax classifier. In forward propagation, GRL acts as an identity transformation. In back propagation, GRL can change the sign of gradient by multiplying -λ. λ can be computed by following formula
(11)$$\lambda =\frac{2}{1+exp(-10\times p)}-1\;,$$
where *p* is a hyperparameter with value from 0 to 1. This strategy allows the domain classifier to be less sensitive to noisy at the early stages of the training procedure. The domain classifier takes feature vectors *Z* as input and outputs
(12)$$G_{d}\left(Z\right)=softmax(relu\left(FC\left(Z\right)\right)$$
where FC(·) denotes the fully connected layer. The softmax classifier contains two output units to denote the target cell type and source cell types, respectively.

#### 4.5.4. Objective Function

We use cross entropy as our loss function. Formally, we use $G_{f}\left(\cdot ;\theta_{f}\right)$, $G_{l}\left(\cdot ;\theta_{l}\right)$ and $G_{d}\left(\cdot ;\theta_{d}\right)$ to denote the feature extractor, the label predictor, and the domain classifier, respectively, where $\theta_{f}$, $\theta_{l}$ and $\theta_{d}$ are their parameters. The prediction loss and the domain loss can be expressed as follows
(13)$$L_{l}\left(\theta_{f},\theta_{l}\right)=L_{l}\left(G_{l}\left(G_{f}\left(x;\theta_{f}\right);\theta_{l}\right),L\right)$$
(14)$$L_{d}\left(\theta_{f},\theta_{d}\right)=-\lambda \times L_{d}\left(G_{d}\left(G_{f}\left(x;\theta_{f}\right);\theta_{d}\right),D\right)$$
where *x* denotes an input DNA sequence, *L* and *D* denote the true label and the true domain of the input DNA sequence *x*, respectively, λ is calculated by Formula (10). The reason that the domain loss is multiply by a negative value is that DANN_TF can help learn features shared by the target cell type and source cell types by decreasing the performance of the domain classifier. Then, learned features can aim to make predictions for TFs in the target cell type.

The total loss L(θ) consists of two parts: the prediction loss and the domain loss. Thus, the total loss function L(θ) can be formulated as
(15)$$L\left(\theta \right)=L_{l}\left(\theta_{f},\theta_{l}\right)+\alpha \times L_{d}\left(\theta_{f},\theta_{d}\right)$$
where α is the domain adaptation parameter.

## Figures and Tables

**Figure 1 ijms-20-03425-f001:**
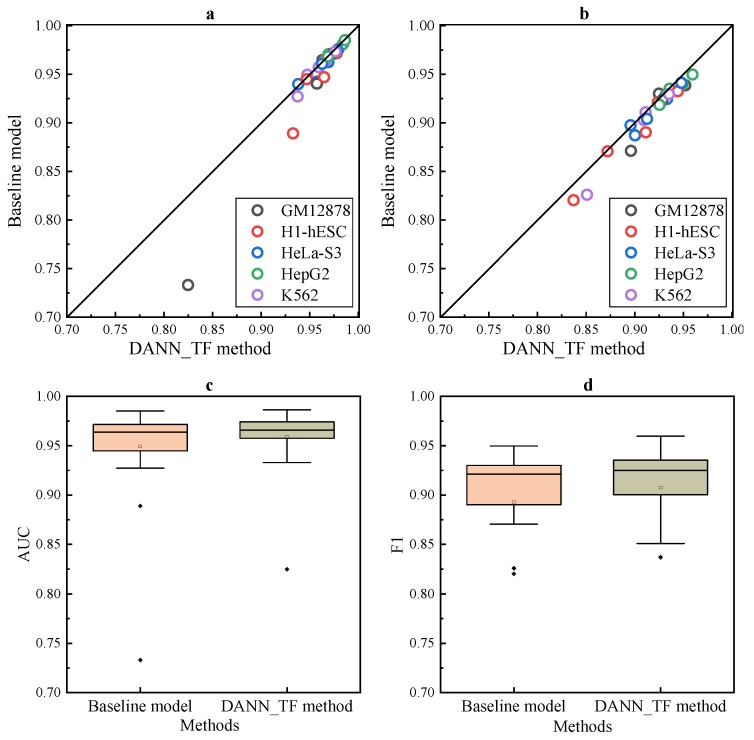
Performance for data augmentation. Scatter plot depicting (**a**) AUC and (**b**) F1 score of DANN_TF and the baseline method. Box plots depicting (**c**) AUC and (**d**) F1 score of DANN_TF and the baseline method.

**Figure 2 ijms-20-03425-f002:**
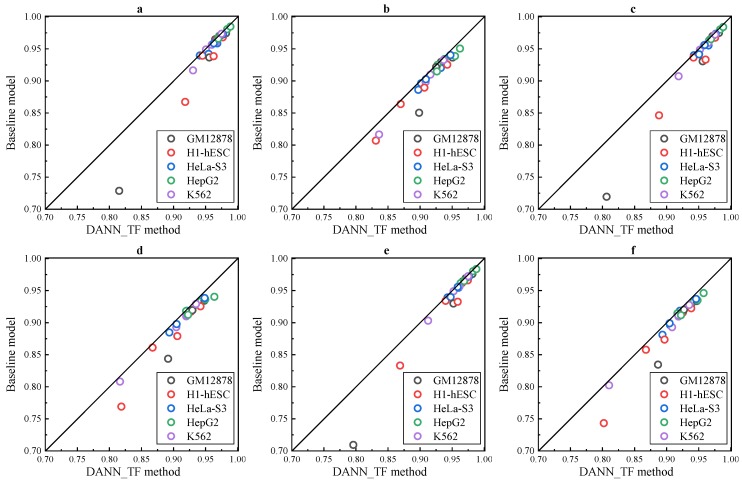
Performance for semi-supervised prediction. AUC and F1 score of DANN_TF and the baseline method for (**a**,**b**) predictions with 50% labeled training data, (**c**,**d**) predictions with 20% labeled training data and (**e**,**f**) predictions with 10% labeled training data.

**Figure 3 ijms-20-03425-f003:**
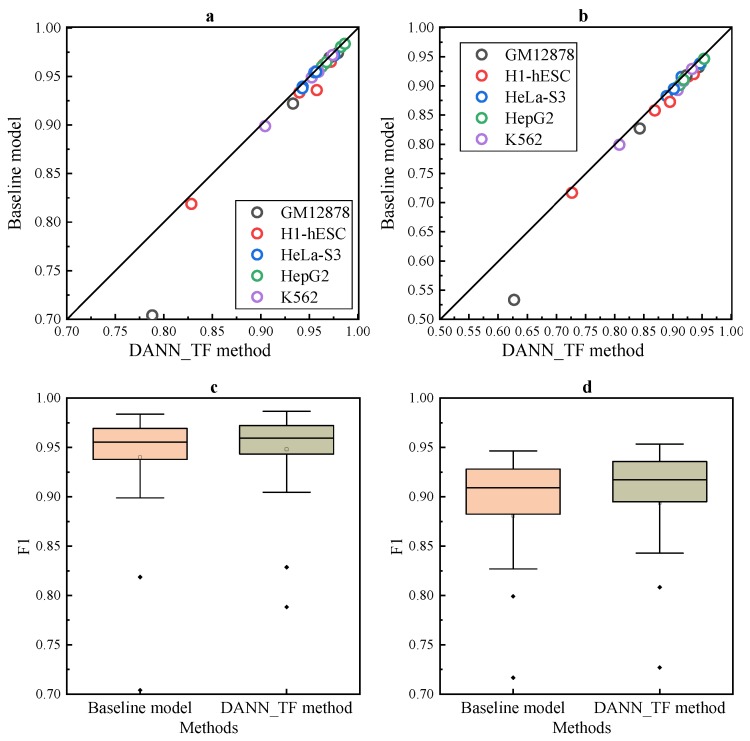
Performance for cross-cell-type prediction. Scatter plot depicting (**a**) AUC and (**b**) F1 score of DANN_TF and the baseline method. Box plots depicting (**c**) AUC and (**d**) F1 score of DANN_TF and the baseline method.

**Figure 4 ijms-20-03425-f004:**
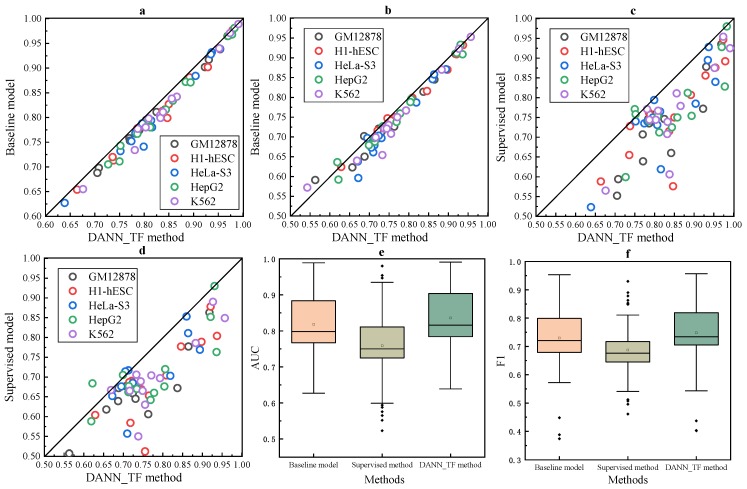
AUCs of cross-cell-type prediction by DANN_TF and the baseline method and that of supervised prediction by the baseline method. Scatter plots depicting the AUC comparison between (**a**) DANN_TF and the baseline method and that (**b**) between DANN_TF and supervised prediction. Scatter plots depicting the F1 comparison between (**c**) DANN_TF and the baseline method and that between (**d**) DANN_TF and supervised prediction. Box plots depicting (**e**) the AUC comparison and (**f**) the F1 comparison between DANN_TF, the baseline method and supervised prediction.

**Figure 5 ijms-20-03425-f005:**
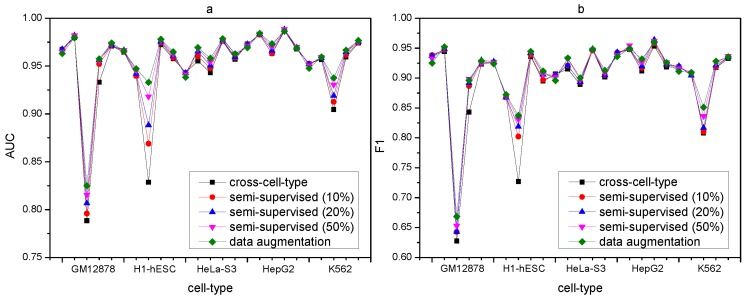
(**a**,**b**) denote AUC performance and F1 score comparison between data augmentation, semi-supervised, and cross-cell-type prediction, respectively.

**Figure 6 ijms-20-03425-f006:**
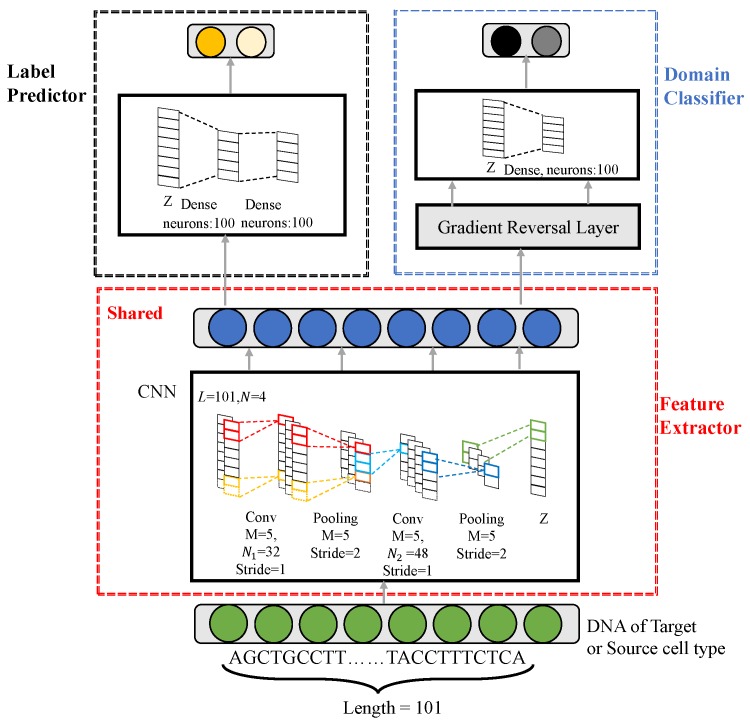
The framework of DANN_TF.

**Table 1 ijms-20-03425-t001:** Performance of DANN_TF for cross-cell-type prediction and that of the baseline method for supervised prediction. (The best performers are marked by bold).

TF	Cell Type	AUC		F1	
		DANN_TF	Baseline	DANN_TF	Baseline
CTCF	GM12878	**0.9631**	0.9419	**0.9250**	0.8819
	H1-hESC	**0.9657**	0.9549	**0.9239**	0.8963
	HeLa-S3	**0.9383**	0.9135	**0.8959**	0.8561
	HepG2	**0.9693**	0.9521	**0.9360**	0.8976
	K562	**0.9476**	0.9227	**0.9113**	0.8676
GABPA	GM12878	**0.9797**	0.9010	**0.9521**	0.8176
	H1-hESC	**0.9472**	0.9278	**0.8721**	0.8369
	HeLa-S3	**0.9692**	0.9394	**0.9334**	0.8786
	HepG2	**0.9842**	0.9352	**0.9492**	0.8587
	K562	**0.9594**	0.9028	**0.9093**	0.8224
JunD	GM12878	**0.8249**	0.6707	**0.6686**	0.3306
	H1-hESC	**0.9329**	0.9030	0.8371	**0.8681**
	HeLa-S3	**0.9581**	0.9156	**0.9003**	0.8654
	HepG2	0.9733	**0.9735**	0.9317	**0.9318**
	K562	**0.9376**	0.9362	0.8510	**0.8609**
REST	GM12878	**0.9575**	0.7826	**0.8961**	0.6674
	H1-hESC	**0.9780**	0.8799	**0.9444**	0.7864
	HeLa-S3	**0.9785**	0.9156	**0.9481**	0.8386
	HepG2	**0.9864**	0.7691	**0.9595**	0.6810
	K562	**0.9666**	0.9504	**0.9282**	0.9017
USF2	GM12878	**0.9741**	0.9607	**0.9287**	0.9211
	H1-hESC	**0.9649**	0.9546	**0.9114**	0.9062
	HeLa-S3	**0.9628**	0.9471	**0.9127**	0.8902
	HepG2	**0.9689**	0.9474	**0.9257**	0.8909
	K562	**0.9767**	0.8069	**0.9353**	0.6958

**Table 2 ijms-20-03425-t002:** Details of the comparison between DANN_TF and the baseline method for cross-cell-type prediction. (Average${}^{a}$ denotes the average improvement, Average${}^{b}$ denotes the micro average over the total number of samples).

Cell Type	GM12878	H1-hESC	HeLa-S3	HepG2	K562	Averageb
Sample total	13	13	13	13	13	13
Improvement total	13	13	13	11	13	12.6
Improvement (%)	100	100	100	84.6	100	96.9
Maximum (%)	2.6	4.8	5.8	3.9	4.7	4.4
Average${}^{a}$ (%)	1.5	1.8	2.0	1.8	2	1.8

**Table 3 ijms-20-03425-t003:** Details of the comparison between cross-cell-type prediction by DANN_TF and supervised prediction by the baseline method. (Average${}^{a}$ denotes the average improvement, Average${}^{b}$ denotes the micro average over the total number of samples).

Cell Type	GM12878	H1-hESC	HeLa-S3	HepG2	K562	Averageb
Sample total	13	13	13	13	13	13
Improvement total	13	13	13	11	13	12.6
Improvement (%)	100	100	100	84.6	100	96.9
Maximum (%)	18.2	27.1	19.7	15.0	23.2	20.6
Average${}^{a}$ (%)	8.8	8.2	6.6	7.82	7.8	7.7

**Table 4 ijms-20-03425-t004:** AUCs of our proposed method and the four state-of-the-art methods. (The best performers and the second-best performers are marked by bold and underline, respectively).

TF	Cell Type	DanQ	DanQ-J	DeepSEA	DeepSEA-J	DANN_TF
CTCF	GM12878	0.780	0.703	0.745	0.617	**0.967**
	H1-hESC	0.824	0.723	0.767	0.656	**0.964**
	HeLa-S3	0.754	0.670	0.699	0.605	**0.943**
	HepG2	0.826	0.724	0.772	0.644	**0.973**
	K562	0.772	0.687	0.720	0.618	**0.952**
GABPA	GM12878	0.929	0.907	0.906	0.895	**0.979**
	H1-hESC	0.922	0.906	0.907	0.894	**0.939**
	HeLa-S3	0.808	0.772	0.766	0.752	**0.955**
	HepG2	0.927	0.914	0.913	0.906	**0.982**
	K562	0.911	0.898	0.900	0.892	**0.956**
JunD	GM12878	**0.835**	0.779	0.789	0.729	0.788
	H1-hESC	0.771	0.718	0.726	0.699	**0.828**
	HeLa-S3	0.850	0.721	0.766	0.671	**0.943**
	HepG2	0.842	0.725	0.765	0.697	**0.963**
	K562	0.717	0.652	0.664	0.624	**0.904**
REST	GM12878	0.750	0.651	0.657	0.630	**0.933**
	H1-hESC	0.699	0.604	0.603	0.580	**0.972**
	HeLa-S3	0.678	0.580	0.584	0.560	**0.975**
	HepG2	0.758	0.673	0.682	0.664	**0.986**
	K562	0.756	0.711	0.715	0.695	**0.959**
USF2	GM12878	0.789	0.706	0.710	0.689	**0.970**
	H1-hESC	0.849	0.775	0.780	0.758	**0.957**
	HeLa-S3	0.723	0.637	0.644	0.609	**0.957**
	HepG2	0.811	0.691	0.693	0.668	**0.967**
	K562	0.809	0.704	0.717	0.691	**0.974**

**Table 5 ijms-20-03425-t005:** The number of TFBSs for the five TFs in the five cell types.

TFs	GM12878	H1-hESC	HeLa-S3	HepG2	K562
CTCF	61,525	104,538	91,807	69,097	80,538
GABPA	4897	16,695	14,309	7883	15,276
JunD	1044	59,352	19,701	90,728	98,704
REST	5503	14,800	12,189	4677	23,655
USF2	30,248	29,972	52,190	27,423	9428

**Table 6 ijms-20-03425-t006:** The number of TFBSs for the additional 13 TFs in the five cell types.

TFs	GM12878	H1-hESC	HeLa-S3	HepG2	K562
CEBPB	11,572	31,114	122,008	113,258	77,430
CHD2	31,194	13,698	41,000	10,338	15,594
EZH2	4944	12,740	3636	6572	3370
MAX	25,084	22,258	59,294	23,708	62,872
MXI1	35,470	12,702	24,018	40,742	13,422
MYC	7380	2434	5670	8826	10,046
NRF1	11,366	9026	5830	3804	8422
NRSF	13,812	26,572	20,494	12,048	31,698
P300	10,336	17,868	51,708	55,826	51,762
RAD21	80,038	111,348	86,840	74,466	35,254
RFX5	8682	3390	38,568	12,034	4402
TAF1	28,556	41,094	32,200	33,318	30,492
TBP	29,786	34,388	36,978	27,612	35,116

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
