# Peer review of "Cross-Cell-Type Prediction of TF-Binding Site by Integrating Convolutional Neural Network and Adversarial Network"

_ijms, 2019, doi:10.3390/ijms20143425_

Reviewer 1 Report

The authors present a method for classifying negative and positive binding sequences. This is an important problem for understanding biology and mechanism of TF-bsed gene regulation. I am missing a lot of the details of how method is implemented and details of training. In addition, the comparisons are not made fairly. I list my comments below:

. Figure 1 is not clear. The circles do not define anything clearly. Also the label is not clear. Please explain in detail what the input and output is on the figure. An example input data on the figure would be very useful.

. Input/output to the method: I cannot understand what the input and output to the algorithm are. From Fig 1, it seems like the sequences from two cell lines are input. But then how is the algorithm unsupervised? In section 4, "When the target cell-type has nolabeled data, DANN_TF amounts to a cross-cell-type method due to the fact that labeled data in the source cell-types are used to predict TFBSs for the target cell-type." but then what is input for the target cell type? What is the "training data of the target cell-type"?

. The accuracy estimates will be affected by the choice of negatives. The impact of selecting negatives must be tested by using different negative sequence generation and plotting the accuracy as in Fig 2.

. Baseline method is not clearly defined. In Figure 2, the baseline and DANN_TF's performance are very similar. I am suspecting that this is because it is fairly straightforward to classify between read and shuffled sequences.

. Between lines 245 and 250, authors refer to maximum improvements in accuracy when DANN_TF and baseline method. Which dataset yields highest increase in accuracy and please explain why this is the case.

. Three of the 5 cell types are cancer cell lines. How do authors justify these as regular cell lines?

. Following is wrong: "For each obtained peak, the midpoint was used 89 as the center to define a TFBS." Please either fix this claim or provide citation for this.

. How can DANN_TF perform better compared to all methods? Especially it is stated that DANN_TF is running with cross-cell type mode?

. Please have the manuscript proof-read.

. Line 33: Fix "position weight matrixs".

. Do not put a dash in "cell-type". Replace it with "cell type". I mean replace "cell-type" with "cell type"

. "However, many TFs of 3 cell-types lack of labeled data.": Not clear. TF's are not of cell-types.

. Not clear: "DANN_TF is further evaluated by additional 13 TFs of the five cell-types and 12 results show that DANN_TF achieves higher AUC than the baseline method and even supervised 13 prediction significantly on 92.3% pairs."

. Major Comment: In the comparisons, it is stated that "The performance of these four state-of-the-art methods were achieved by using their default setup and parameters". The other methods must be re-trained with the training data that is used in the paper otherwise it will not be fair to them. Authors must re-train those methods on the training data used with DANN_TF and test them on the target data.

. The details of the training algorithm are not described for CNN and the GAN method. What are the specific parameters (batch size, dropout rate, network architecture, etc)? Which package is used for implementation?

Author Response

Response to Reviewer 1 Comments

Comments to the Author 

The authors present a method for classifying negative and positive binding sequences. This is an important problem for understanding biology and mechanism of TF-bsed gene regulation. I am missing a lot of the details of how method is implemented and details of training. In addition, the comparisons are not made fairly. I list my comments below:

Main suggestions: 

Point 1. Figure 1 is not clear. The circles do not define anything clearly. Also the label is not clear. Please explain in detail what the input and output is on the figure. An example input data on the figure would be very useful.

Response 1: Thank you very much first for your overall detailed review and suggestions. Based on your suggestion, we redrawn Figure 1. The current Figure 1 shows the three components of our proposed DANN_TF and their detailed structures. The green circles denote a input sequence, the blue circles denote learned features by the feature extractor, the dark yellow circle and the pale yellow circle denote TFBS and non-TFBS, respectively, and the dark black circle and the pale black denote the target cell type and the source cell types, respectively. The input to DANN_TF is a sample of the target cell-type or the source cell types. For example, when we use DANN_TF to predict TFBSs of CTCF in GM1278, then GM12878 is the target cell type while the other four cell types are the source cell types. Then the input to DANN_TF is a sample of GM12878 or the other four cell types. Figure 1 also provides an example input sequence. The output contains two components: the Label Predictor is used to predict the input sequence into TFBS or non-TFBS while the Domain Classifier is used to predict whether the input sequence is belong to the target cell type or the source cell types. The revised text is shown in red in line 142-153.

Point 2. Input/output to the method: I cannot understand what the input and output to the algorithm are. From Fig 1, it seems like the sequences from two cell lines are input. But then how is the algorithm unsupervised? In section 4, "When the target cell-type has no labeled data, DANN_TF amounts to a cross-cell-type method due to the fact that labeled data in the source cell-types are used to predict TFBSs for the target cell-type." but then what is input for the target cell type? What is the "training data of the target cell-type"?

Response 2: Based on your suggestion, we redrawn Figure 1. Current Figure 1 shows that the input to DANN_TF is an DNA sequence, which can be a sample of the target TF in the target cell type or the source cell types. The output contains two components: the Label Predictor is used to predict the input sequence into TFBS or non-TFBS while the Domain Classifier is used to predict whether the input sequence is belong to the target cell type or the source cell types.

As our goal is to predict TFBSs of TFs in the target cell type, if all the training data in the target cell type is unlabeled, then DANN_TF is trained by the unlabeled training data of the target cell type and the labeled training data of the source cell types, thus DANN_TF is an unsupervised method. For example, when we use DANN_TF to predict TFBSs of CTCF in GM1278, then GM12878 is the target cell type, and the other four cell types are the source cell types. If the training data of CTCF in GM12878 is unlabeled, then the input for the target cell type are the unlabeled training data of CTCF in GM12878. In this case, the training data of DANN_TF contains the unlabeled training data of CTCF in GM12878 and labeled training data of CTCT in other four cell types. As DANN_TF do not use any labeled training data of GM12878, predictions of DANN_TF for CTCF in GM12878 are unsupervised predictions. The revised text is shown in red in line 329-335.

Point 3. The accuracy estimates will be affected by the choice of negatives. The impact of selecting negatives must be tested by using different negative sequence generation and plotting the accuracy as in Fig 2.

Response 3: Based to your suggestion, we evaluate effects of two negative sequence generation methods on the performance of our proposed DANN_TF. Results show that the performance of the shuffle method is only 0.1 percentage higher than that of the random method. Therefore, different negative sequence generation methods have little effects on the performance of our proposed DANN_TF. The added text is shown in red in line 247-261.

Point 4. Baseline method is not clearly defined. In Figure 2, the baseline and DANN_TF's performance are very similar. I am suspecting that this is because it is fairly straightforward to classify between read and shuffled sequences.

Response 4: DANN_TF combines the use of CNN and Adversarial Network to learn common features among the target cell type and the source cell types while the baseline method is similar to DANN_TF except that it does not use Adversarial Network in prediction. One possible reason that the baseline and DANN_TF's performance are very similar is that DANN_TF achieves lower AUC than the baseline method for CTCF in four cell types, because labeled training data of CTCF in these four cell types are sufficient and augmenting their training data by labeled data available in other cell types may bring a lot of noise. The added text is shown in red in line 288-291.

Point 5. Between lines 245 and 250, authors refer to maximum improvements in accuracy when DANN_TF and baseline method. Which dataset yields highest increase in accuracy and please explain why this is the case.

Response 5: Based on your question, we provided the datasets on which our proposed DANN_TF achieves the highest improvement in accuracy and explained the possible reason. DANN_TF achieves the maximum performance improvement when predicting JunD in GM12878. One possible reason is that labeled training data of JunD in GM12878 is less than other cases and DANN_TF can leverage labeled data of JunD available in the source cell types to predict its TFBSs in GM12878. The revised text is shown in red in line 277-280. 

Point 6. Three of the 5 cell types are cancer cell lines. How do authors justify these as regular cell lines?

Response 6: We deleted the inappropriate descriptions in our manucript. HeLa-S3, HepG2 and K562 are three cancer cell lines. As they have been widely studied by many researchers, TFBSs of many TFs in these three cell lines have been identified by experimental methods, such as Chip-chip and ChiP-seq. So these three cell lines contain known TFBSs for enough number of TFs which can be used to evaluate the performance of our proposed DANN_TF. As GM12878 and H1-hESC are regular cell lines while HeLa-S3, HepG2 and K562 are three cancer cell lines, their combinations can further validate that our proposed DANN_TF trained by regular cell lines can be used to predict TFBSs accurately for cancer lines and vice verse. 

Point 7. Following is wrong: "For each obtained peak, the midpoint was used as the center to define a TFBS." Please either fix this claim or provide citation for this.

Response 7: Based on your suggestion, we provided citations for this sentence. The revised text is shown in red in line 86-88.

Point 8. How can DANN_TF perform better compared to all methods? Especially it is stated that DANN_TF is running with cross-cell type mode?

Response 8: The baseline method is similar to DANN_TF except that it does not use Adversarial Network in prediction. For data augmentation, our proposed DANN_TF and the baseline method are trained by the labeled training data of the target cell type and all labeled data of the source cell types. Results show that Adversarial Network can learn common features among the target cell type and the source cell types. Therefore, DANN_TF perform better than the baseline method. For the semi-supervised prediction, to evaluate the influence of the number of labeled training data of the target cell type on DANN_TF, we suppose that only a portion of training data of the target cell type is labeled (50%, 20%, 10%). Results show that, when the proportion of the labeled training data decreases, the performance of the baseline method drops a lot while the performance of DANN_TF drops a little. For cross-cell-type prediction, we suppose that all the training data of the target cell type is unlabeled while the validation data as well as the test data are labeled. Thus, our proposed DANN_TF is trained by combining the unlabeled training data of the target cell type and labeled data of the source cell types, whereas the baseline method can only be trained by labeled data of the source cell types. Results indicate that our proposed DANN_TF can achieve better performance than the baseline method. It indicates that DANN_TF indeed can learn common features among the target type and the source cell types to improve the performance for cross-cell-type prediction.

Point 9: Please have the manuscript proof-read.

Response 9: 1 Line 33: Fix "position weight matrixs".

The statement is revised, which is shown in red in the line 31.

2 Do not put a dash in "cell-type". Replace it with "cell type". I mean replace "cell-type" with "cell type"

Based on your suggestion, we replaced all "cell-type" with "cell type"

3 However, many TFs of 3 cell-types lack of labeled data.": Not clear. TF's are not of cell-types.

We revised this sentence into “However, many TFs lack of labeled data in a lot of cell types” and we also revised all other “TFs of cell types” in our manuscript.

 4. Not clear: "DANN_TF is further evaluated by additional 13 TFs of the five cell-types and results show that DANN_TF achieves higher AUC than the baseline method and even supervised prediction significantly on 92.3% pairs."

We revised this sentence into “DANN_TF is further evaluated by additional 13 TFs in the five cell types with a total of 65 cell type TF pairs. Results show that DANN_TF achieves higher AUC than the baseline method significantly on 96.9 % pairs of the 65 cell type TF pairs.” The revised text is shown in red in line 10-12.

Point 10. Major Comment: In the comparisons, it is stated that "The performance of these four state-of-the-art methods were achieved by using their default setup and parameters". The other methods must be re-trained with the training data that is used in the paper otherwise it will not be fair to them. Authors must re-train those methods on the training data used with DANN_TF and test them on the target data.

Response 10: Thank you for your positive suggestions. In fact, the four state-of-the-art methods were indeed trained with the training data that is used by our proposed DANN_TF by using their default setup and parameters. I'm sorry we didn't write it clearly in the original manuscript. We revised the manuscript to clearly demonstrate the training and the evaluation of these four state-of-the-art methods and their comparison with our proposed DANN_TF. The revised text is shown in red in line 430-433. 

Point 11. The details of the training algorithm are not described for CNN and the GAN method. What are the specific parameters (batch size, dropout rate, network architecture, etc)? Which package is used for implementation?

Response 11: The input of the feature extractor is an feature matrix with dimension of N×L=4×101. The feature extractor in DANN_TF uses two convolution layers with N_1=32 and N_2=48 kernels, respectively, and each is followed by a max pooling layer. The kernel size of the two convolution layers and the pooling size of the two pooling layers are M=5. The stride of the two convolution layers and the two pooling layers are 1 and 2, respectively. A dropout regularization layer with dropout probability of 0.7 is used to avoid overfitting. The label predictor has two fully connected layers of 100 neurons followed by a softmax classifier. The domain classifier has a fully connected layer with 100 neurons followed by a softmax classifier. The domain adaptation parameter α is set to 1. Our DANN_TF model was trained using the Momentum algorithm with batch size of 128 instances. The momentum and the learning rate are set as 0.9 and 1e-3, respectively. DANN_TF was written in Python and implemented using the tensorflow-gpu 1.0 library. The revised text is shown in red in line 222-232.

Reviewer 2 Report

The paper is well written with good enough background and nicely explained results.

I have the following questions and comments for the authors:

1) in the abstract, expand the first reference of TF

2) probably if the authors try positional encoding along with one hot encoding, it might capture positional correlations in the embedding as well.

3) Ln 123: please provide the size of the encoded vector and the given vector was confusing. 

4) Ln 152: It will be good to give the entire network graph for your ML workflow.

6. Please provide standard dimensions for N, M, L, K, C so on.

7. Why is the domain loss negative?

8. Ln 240. seems like you used the labeled target types both in training and testing? You should not use the same data for the test and train!

9. Try to explain why sometimes the baseline model is better? Are you overfitting?

10. Also comment on runtime of baseline, DANN_TF, and other state-of-the-arts?

11. Ln: 372. Can you scientifically explain why baseline is better?

Author Response

Response to Reviewer 2 Comments

Comments to the Author :

The paper is well written with good enough background and nicely explained results.

Point 1. in the abstract, expand the first reference of TF

 Response 1: Thank you very much first for your overall detailed review and suggestions. We revised our abstract and expanded the first reference of TF. The revised text is shown in red in line 1-3.

 Point 2. probably if the authors try positional encoding along with one hot encoding, it might capture positional correlations in the embedding as well.

 Response 2: Based on you suggestion, we tried to encode DNA sequences by combining a positional encoding method and the one-hot encoding method. In this work, we use sine and cosine functions of different frequencies [1] to encode positional features:

where pos is the position and i is the dimension. Each dimension of the positional encoding corresponds to a sinusoid. The wavelengths form a geometric progression from  to .

The positional encoding and the one-hot encoding are integrated by Vector Addition. JunD in the five cell types are taken as examples to evaluate the effect of the positional encoding on the performance of TFBS prediction. Results of DANN_TF and the baseline method with or without the positional encoding on JunD in the five cell types are shown in following table. Results show that the position encoding has a negative impact on both DANN_TF and the baseline method.

Cell type

DANN_TF

Baseline

DANN_TF_pos

Baseline_pos

GM1278

0.7883

0.7039

0.783

0.693

H1-hESC

0.8286

0.8187

0.829

0.810

HeLa-S3

0.9430

0.9380

0.938

0.934

HepG2

0.9630

0.9600

0.958

0.945

K562

0.9046

0.8989

0.882

0.875

DANN_TF_pos and Baseline_pos denotes DANN_TF+position and Baseline+position, respectively.

[1] Jonas Gehring, Michael Auli, David Grangier, Denis Yarats, and Yann N. Dauphin. Convolutional sequence to sequence learning. arXiv preprint arXiv:1705.03122v2, 2017.

 Point 3. Ln 123: please provide the size of the encoded vector and the given vector was confusing.

 Response 3: DNA is composed by four nucleotide types (A, G, C, T), so the dimension of one-hot vectors of the four nucleotide types is four. As both TFBSs and non-TFBSs are composed by 101 nucleotides, their encoded vectors are matrices of dimension . We have revised our manuscript. The revised text is shown in red in line 116-117.

Point 4. Ln 152: It will be good to give the entire network graph for your ML workflow.

 Response 4: Based on your suggestion, we redrawn Figure 1 to provide the entire network of our proposed DANN_TF. Details of the entire network graph is also provided in the text. The revised text is shown in red in line 142-153.

Point 5. Please provide standard dimensions for N, M, L, K, C so on.

 Response 5: Based on your suggestions, the values of these hyperparameters is provided in the current Figure 1. We also added the section 4.1 into the manuscript to introduce the values of these hyperparameters. The revised text is shown in red in line 222-232.

 Point 6. Why is the domain loss negative?

 Response 6: We changed “-“ to “+“. Domain classifier contains a Gradient Reversal layer (GRL). In forward propagation, GRL acts as an identity transformation. In back propagation, GRL can change the sign of gradient by multiplying -. So, when DANN_TF is backpropagating, we get a negative loss. The reason that the domain loss is multiply by a negative value is that DANN_TF can help learn shared features of the target cell types and the source cell types by decreasing the classification performance of the domain classifier. The revised text is shown in red in line 197-199.

Point 7. Ln 240. seems like you used the labeled target types both in training and testing? You should not use the same data for the test and train!

Response 7: For each TF in each cell-type, the labeled data are divided into 10 separate folds of equal size: 8 folds for training, 1 fold for validation and 1 fold for test. The above process is repeated for ten times until each fold is tested once. Finally, the performance on the 10 folds are averaged. To evaluate the performance of data augmentation by DANN_TF, our proposed DANN_TF and the baseline method are trained by the training data of the target cell type and all labeled data of the source cell types. They are validated and tested by the validation data and the test data of the target cell type, respectively. The revised text is shown in red in line 218-221 and line 263-266.

Point 8. Try to explain why sometimes the baseline model is better? Are you overfitting?

 Response 8: DANN_TF combines the use of CNN and Adversarial Network to learn common features among the target cell type and the source cell types while the baseline method is similar to DANN_TF except that it does not use Adversarial Network in prediction. It is found that sometimes the baseline model is better than our proposed DANN_TF, such as CTCF in four cell types. One possible reason is that labeled training data of CTCF in these four cell types are sufficient and augmenting their training data by labeled data available in other cell types may bring a lot of noise. The revised text is shown in red in line 288-291.

 Point 9. Also comment on runtime of baseline, DANN_TF, and other state-of-the-arts?

 Response 9: Based on your suggestion, we discussed the runtime of the baseline method, DANN_TF and the four state-of-the-art methods. We compared the training time of DANN_TF for each epoch to that of the baseline method and the four state-of-the-art methods. JunD in the five cell types are taken as examples to evaluate their efficiency. All the methods are trained on NVIDIA GeForce RTX 2080Ti. Results show that DANN_TF takes 45s, 42s, 44s, 42s and 45s for GM12878, H1-hESC, HeLa-S3, HepG2 and K562, respectively, and the training time of the baseline method for these five cell types are 33s, 34s, 32s, 28s and 29s, respectively. The training time of DanQ, DanQ-JASPAR, DeepSEA and DeepSEA-JASPAR are 401s, 355s, 96s and 85s, respectively. As their predictions for JunD in the five cell types are completed by a single model, their average training time for each cell type are 80s, 71s, 19s and 17s, respectively. In summary, DANN_TF takes less time than DanQ and DanQ-JASPAR and spent a little more time than the baseline method, DeepSEA and DeepSEA-JASPAR. The revised text is shown in red in line 438-447.

 Point 10. Can you scientifically explain why baseline is better?

 Response 10: DANN_TF combines the use of CNN and Adversarial Network to learn common features among the target cell type and the source cell types while the baseline method is similar to DANN_TF except that it does not use Adversarial Network in prediction. It is found that sometimes the baseline model is better than our proposed DANN_TF, for example CTCF in four cell types. One possible reason is that labeled training data of CTCF in these four cell types are sufficient and augmenting their training data by labeled data available in other cell types may bring a lot of noise. The revised text is shown in red in line 288-291.

Round  2

Reviewer 1 Report

Authors have done a fair job in the revision to clarify the presentation and update the algorithm's details. I appreciate authors efforts. I have the following comment before I can recommend manuscript to be accepted for publication:

1. DANN_TF assumes sequence specificity of transcription factors. For example CTCF and Jun have well-studied sequence motifs. Please make a test run on DANN_TF using TFs that do not show significant sequence specific binding. I think this will be useful for discussing limitations of DANN_TF algorithm.

Author Response

Response to Reviewer 1 Comments (Round 2) Comments to the Author : Authors have done a fair job in the revision to clarify the presentation and update the algorithm's details. I appreciate authors efforts. I have the following comment before I can recommend manuscript to be accepted for publication: Point 1. DANN_TF assumes sequence specificity of transcription factors. For example, CTCF and Jun have well-studied sequence motifs. Please make a test run on DANN_TF using TFs that do not show significant sequence specific binding. I think this will be useful for discussing limitations of DANN_TF algorithm. Response 1: Thank you very much first for your overall detailed review and suggestions. Among the 13 TFs in the second data set, eight TFs are sequence-specific and have specific binding motifs. The remaining five TFs (CHD2, EZH2, NRSF, RAD21 and TAF1) do not have specific binding motifs in any of the three common databases (JASPAR, TRANSFAC and Uniprobe). As these TFs do not have specific binding motifs which can be learned by CNN from labeled data in cross-cell-type, they may achieve low improvements than the sequence-specific TFs. Results listed in Supplementary Table S7 show that DANN_TF achieves higher AUC than the baseline method and supervised prediction by 1.6% and 7.2% on average, respectively, for the TFs without specific binding motifs. For the sequence-specific TFs, DANN_TF achieves higher AUC than the baseline method and supervised prediction by 2.0% and 8.0% on average, respectively. Although DANN_TF achieves lower improvements for the TFs without specific binding motifs than the sequence-specific TFs, DANN_TF also achieves prominent improvements for them. It indicates that DANN_TF can learn binding features from labeled data in cross-cell-type for the TFs which do not have specific binding motifs. The revised text is shown in red in line 108-111 and line 416-426.
